# Intrusion errors during verbal fluency task in amyotrophic lateral sclerosis

**Manuel Perez** [1] *, **Imanol Amayra**[1], **Esther Lazaro**[1], **Maitane García**[1], **Oscar Martínez**[1], **Patricia Caballero**[1,2], **Sarah Berrocoso**[1], **Juan Francisco López-Paz**[1], **Mohammad Al-Rashaida**[1], **Alicia Aurora Rodríguez**[1], **Paula Luna**[1], **Luis Varona**[3]

**1** University of Deusto, Vizcaya, Spain, **2** Clinical Psychology, Galdakao University Hospital, Vizcaya, Spain, **3** Department of Neurology, Basurto University Hospital, Vizcaya, Spain

* manuel.perez@deusto.es

## Abstract

### Background

Numerous studies have noted the presence of a dysexecutive component of the ALS-FTD. The most widely replicated result refers to the significantly reduced verbal fluency of ALS patients when compared to healthy people. As ALS patients have motor alterations that interfere with production, qualitative studies have the advantage of being independent of the degree of motor disability and revealing patients' cognitive state. This study examined the production differences between 42 ALS patients who presented with different degrees of dementia and motor impairment and 42 healthy people. Production processes were studied by extending the administration time of a letter fluency task to 2 minutes for the phonemic verbal fluency (PVF) and semantic verbal fluency (SVF) categories. This ensured that the qualitative aspects of verbal fluency were addressed, paying special attention to the new perseverations and intrusions, as well as any clinical correlates that may exist.

### Results

The ALS patients produced a significantly lower number of responses in PVF ($p = .017$) and SVF ($p = .008$). The rest of the indicators for frontal lobe alteration also suggested the existence of a dysfunction. The most remarkable results were the number of intrusions on the PVF task, which was much higher in the ALS group ($p = .002$). However, the number of perseverations did not differ significantly.

### Conclusions

This study highlights the value of intrusions in addressing cognitive deterioration in ALS patients. This deterioration seems to be independent of the degree of motor impairment and of behavioural alterations. Therefore, the value of the intromissions on the verbal fluency task was highlighted as an indicator of a new cognitive alteration, which can be easily evaluated, even retrospectively.

**Data Availability Statement:** All relevant data are within the paper and its Supporting Information files.

**Funding:** The author(s) received no specific funding for this work.

**Competing interests:** The authors have declared that no competing interests exist.

## Introduction

ALS is a progressive neurodegenerative disease that produces muscle weakness and flaccidity, fasciculations and spasticity, in combination with breathing and swallowing problems [1]. It is a rare disease that has an incidence of 1–2.6 cases per 100 00 people/year, with an approximate prevalence of 6 cases per 100,000. Death usually occurs within 2 to 3 years from the onset of symptoms [2, 3].

About 3% -5% of patients with ALS are diagnosed with frontotemporal dementia (FTD) [4, 5]. Most cases of ALS-FTD present the behavioural variant of the disease, although there may also be a predominant impairment of language (i.e. primary progressive aphasia, or semantic dementia) [6, 7]. ALS-FTD usually presents with symptoms such as apathy, disinhibition, loss of empathy and repetitive behaviour [8].

In addition to behavioural changes, about 30% of ALS patients experience cognitive impairment [9], in areas such as language [10], memory [11], visuospatial function [12], and Theory of Mind [13]. However, the deterioration of executive functions is the most consistent finding [14]. While motor dysfunction is usually present in ALS patients, the most severe deficiencies are found in verbal fluency tasks [15, 16]. The verbal fluency task requires initiation processes, strategy formation, sustained attention, attentional changes, inhibition and working memory [17]. In the standard version of the task, participants are given 1 minute to produce as many words as they can beginning with a certain letter (phonetic fluency), or to name as many words as possible within a given category (semantic fluency). The participant's score is the correct number of words generated [18–20]. However, it has been noted that extending the letter fluency task to 2 minutes can increase sensitivity to detect cognitive alterations [21].

The significant decrease in the number of words produced by patients with ALS led to the use of different approaches to understand the mechanisms involved in the generation of words. The strategies used by researchers such as Troyer and her colleagues may be of interest [22, 23]. These authors separated the compounds of verbal fluency into two main processes: clustering and switching. Clustering is the accumulation of words and is based on the storage of verbal memory, while switching is the process of changing from one group to another, and is based on the function of the frontal lobe. Switching relies on the executive functions, as change and the use of strategies are required [24]. Troyer's qualitative approach has been used previously in ALS and it has been shown that patients have a significantly decreased performance, caused by frontal degeneration [25–27]. This approach has not been widely used but is of interest due to its independence from motor impairment, which can reduce the number of words generated.

Following the quality of the responses produced during the evaluation of verbal fluency, error patterns can be found, based on perseverations and intrusions. A perseveration error is the repetition of a previous response [28], while intrusion errors include all the words produced that do not belong to the required semantic field or do not begin with the required letter [29]. The appearance of these errors may suggest the presence of executive deterioration and has been described in patients with FTD [30], schizophrenia [31], Parkinson's disease [32], brain injury [33], Alzheimer's disease [34–37], and progressive supranuclear palsy [38]. In most studies, intrusions have been studied by means of a verbal retrieval task.

To our knowledge, the presence and value of intrusions and perseverations within the verbal fluency task have not been previously evaluated in ALS. Given the emphasis on frontal dysfunction in ALS-FTD, the objective of this study is to evaluate the usefulness of a verbal fluency response test in a sample of ALS, and to compare the results with the healthy population, as well as with the related clinical variables. According to the hypothesis posed here, patients with ALS will be less able to inhibit irrelevant information, so they will produce more

intrusion and perseverance errors. In addition, it is held that these errors will not be related in a statistically significant way to the degree of motor alteration, but rather to a greater severity of behavioural changes.

## Methods

Forty-two patients diagnosed with definite or probable ALS according to the El Escorial criteria for [39] were recruited from the Motor Neuron Disease Units. In addition, forty-two healthy volunteers were recruited, matched according to sex, age and years of education. Prior to commencing the neuropsychological evaluations, the purpose of the research was explained to all participants, and then each person was asked to sign the consent form to ensure that they were a willing participant in the research. The informed consent form was approved by the research program's human subjects board. Also, 42 unrelated healthy volunteer controls matched for sex, age and years of education were recruited through public advisements in hospitals. The exclusion criteria for both groups were having suffered brain injury or stroke, psychological problems and/or severe psychiatric disorders, or alcohol/substance abuse, critical levels of anxiety and/or depression at the time of the evaluation based on the HADS cutoff score, in addition to non-acceptance of informed consent. Two volunteers with ALS were excluded due to previous ischemic strokes.

A clinical trial was conducted on the ALS group using the Spanish version of the ALS-Revised Functional Rating Scale (ALSFRS-R) [40], peak cough flow, and other clinical variables such as age of onset, duration of illness, and rate of disease progression. The rate of disease progression is calculated as = (48 –ALSFRS-R at time of diagnosis) / duration from onset to diagnosis (months) [41]. In addition, the Frontal Behaviour Inventory (FBI) [42] was administered. The FBI incorporates symptoms of habitual behaviour of FTD, according to the criteria set by Lund and Manchester [43]. This is a Likert-type questionnaire aimed at carers on 24 categories of negative uninhibited behaviour. This test provides cut-off points to determine the severity of the behavioural alteration (none, mild, moderate and severe) and ensure face and contextual validity, with high inter-rater reliability (Cohen's *kappa* = 0.9) and internal consistency (Cronbach's *alpha* = 0.9) [44]. In addition, phonological verbal fluency (PVF) (letters P, M and R for the Spanish version) and semantic verbal fluency (SVF) (animals) were evaluated. In this case, participants had 2 minutes for each letter in PVF and SVF. For PVF, the use of proper nouns or words that began with a letter different from the one provided was considered to be an intrusion. The qualitative aspects of verbal fluency were evaluated according to the criteria of Troyer et al. [22]. Repetitions and intrusions were excluded if there were clustering and switching. The presence of language disorders was assessed using the Boston Naming Test (BNT) [45]. In addition, the emotional state of all participants was assessed using the Hospital Anxiety and Depression Scale (HADS) [46].

All patients were included after informed written consent forms approved by the research program's human subjects board. Also, caregivers were consulted to determine whether participants were able to consent. The firm of consent was obtained from patients or their caregivers, in case of severe motor impairment. The controls were informed written consent by themselves, as set forth by the Declaration of Helsinki and its subsequent amendments. Full approval for the study was obtained from the Hospital Ethics Committee.

The data were analysed using the statistical package for the social sciences (SPSS), version 20. The assumptions of normality and homogeneity for parametric tests were examined for the variables using the Shapiro-Wilk test. The comparisons between groups were made using the Mann-Whitney and Kruskal-Wallis tests. Spearman correlations and discriminant function analyses were used to examine the relationships between neuropsychological scores and

clinical measures. All standard and raw scores were transformed into z-scores. All the tests were two-tailed and the statistical significance was established at $p = .005$.

## Results

The clinical and control groups did not differ in age, sex and years of education (Table 1).

With respect to neuropsychological measures, the ALS group had significantly poorer performance in the BNT and PVF tests, together with a greater number of intrusions and switching. With respect to SVF, fewer words and fewer clusters were produced (Table 2). Strikingly, in both groups a statistically significant correlation was found between the number of words in PVF and the number of perseverations. However, the greater number of words in the ALS group was related to more perseverations ($r = .414$, $p = .006$), whereas this trend was reversed in the control group ($r = -.376$, $p = .014$). That is, fewer words were related in a statistically significant manner to more perseverations.

Regarding the clinical characteristics of the ALS group, the number of years of education was significantly correlated to the number of intrusions ($r = -.351$, $p = .023$) and PVF switches ($r = .387$, $p = .011$). In addition, motor impairment was related to PVF score ($r = .346$, $p = .025$), and SVF ($r = .375$, $p = .014$), and SVF clusters ($r = .399$, $p = .009$). Furthermore, those participants with bulbar onset did not differ from those with spinal onset in none of the cognitive measures.

Once the statistically significant differences were identified in the non-parametric analysis between the ALS and the control group, a stepwise discriminant function analysis was performed to establish five discriminant variables between the ALS and the control group (see Table 2). The final model indicates that there are two discriminant variables: PVF score and intrusions (Wilk's Lambda = .835, $\chi^2 = 14.626$, p = .001). The classification table indicates that 70.2% of the subjects were correctly classified. In the ALS group, nine patients (21.4%) performed as the control group in the PVF score and in the number of intrusions. A comparative analysis was performed between the subgroup of ALS patient comparing the clinical variables and differences in the age at disease onset ($U = 114.5$, $p = .043$) and in the disease progression rate ($U = 92.5$, $p = .008$) were found.

With respect to behaviour, the total FBI score was significantly correlated to the number of switches produced in PVF ($r = -.324$, $p = .036$) and to their clusters ($r = -.305$, $p = .049$), and to SVF ($r = -.358$, $p = .020$). When applying the cut-off points of the FBI test, those with severe alterations showed a significantly lower number of words in PVF and their switches, as well as

**Table 1. Demographic and clinical characteristics of the ALS and control groups.**

| | ALS group | | Control group | | |
| --- | --- | --- | --- | --- | --- |
| | % | median (IQR) | % | median (IQR) | *p* |
| Sex (m/f) | 62.8/34.9 | - | 62.8/34.9 | - | .590 |
| Age | - | 62 (15) | - | 11.8 (16) | .864 |
| Education (years) | - | 12 (5) | - | 12 (5) | .804 |
| HADS | - | 7 (9) | - | 6.5 (5) | .676 |
| Disease duration (months) | - | 18 (20) | - | - | - |
| Bulbar onset, no. (%) | 23.80 | - | - | - | - |
| ALSFRS-R | - | 36.5 (12) | - | - | - |
| Cough Peak Flow | - | 295 (202.5) | - | - | - |
| Disease progression rate | - | 0.5 (0.67) | - | - | - |

IQR: interquartile range; *p* values from Mann-Whitney *U* test.

**Table 2. Comparison between control group and ALS group on verbal fluency and its components.**

| | ALS group | Control group | | |
| --- | --- | --- | --- | --- |
| | median (IQR) | median (IQR) | 95%CI | p |
| BNT | 25 (7) | 28 (4) | -9.78 to -1.37 | .011* |
| PVF score | 52.5 (37) | 68 (28) | -9.95 to -1.10 | .017* |
| Perseverations | 1 (2) | 1 (2) | -5.53 to 0.31 | .974 |
| Intrusions | 0 (2) | 0 (0) | 0.24 to 1.07 | .002** |
| Switches | 29 (19) | 35 (23) | -0.94 to -1.10 | .036 |
| Clusters | 13 (10) | 14 7() | -0.61 to 0.25 | .377 |
| Cluster size | 5.23 (1.93) | 30.5 (3.09) | -0.67 to 0.20 | .073 |
| SVF score | 25 (13) | 30.5 (13) | -0.96 to -0.12 | .008* |
| Perseverations | 0 (1) | 0 (1) | -0.25 to 0.61 | .688 |
| Switches | 8.5 (4) | 6 (5) | -0.72 to 0.13 | .262 |
| Clusters | 5.5 (3) | 6 (3) | -0.91 to -0.06 | .026* |
| Cluster size | 3 (1.92) | 3.29 (1.77) | -0.47 to 0.39 | .385 |

in the SVF score (Table 3). In addition, those ALS patients without behavioural alterations did not differ in the PVF score from the control group ($U$ = 186, p = .063), neither in the SVF score ($U$ = 170.5, p = .170).

## Discussion

This study was conducted with the purpose of evaluating the qualitative components of verbal fluency in patients with ALS, paying special attention to the errors committed while performing the task. Despite our baseline hypotheses, ALS patients did not differ in a statistically significantly manner in perseverations in non-semantic phonetic modalities. However, the ALS group produced a remarkably higher number of intrusions. In addition, the data suggest that these errors were independent of other clinical characteristics of the disease, such as the severity of motor impairment and the disease onset type, even though they were linked to a more severe progression and a late onset of the disease. In this sense, although some previous studies

**Table 3. Differences in verbal production according to the degree of behavioural impairment.**

| | None (n = 6) | Mild (n = 7) | Moderate (n = 23) | Severe (n = 6) | |
| --- | --- | --- | --- | --- | --- |
| | median (IQR) | median (IQR) | median (IQR) | median (IQR) | p |
| PVF score | 46.5 (24) | 67 (53) | 55 (40) | 28.5 (47) [a] | .038 |
| Perseverations | 1 (4) | 1 (2) | 1 (2) | 0 (1) | ns |
| Intrusions | 1 (2) | 2 (3) [a] | 0 (1) | 0.5 (2) | .004 |
| Switches | 28 (13) | 30 (22) | 31 (19) | 11.5 (0.25) [a] | .044 |
| Clusters | 13.5 (11) | 18 (10) | 13 (12) | 6.5 (13) | ns |
| Cluster size | 5.42 (2.33) | 5.16 (3.36) | 5.42 (1.95) | 4.08 (4.11) | ns |
| SVF score | 24 (16) | 33 (10) | 25 (12) | 20.5 (15) [a] | .007 |
| Perseverations | - [b] | 0 (2) | 0 (1) | 0 (0) | ns |
| Switches | 8 (5) | 11 (6) | 9 (4) | 5 (9) | ns |
| Clusters | 6 (3) | 7 (2) | 5 (3) | 5 (6) | ns |
| Cluster size | 2.45 (3.52) | 3 (1.55) | 3.11 (1.89) | 2.91 (2.94) | ns |

IQR: interquartile range; PVF: phonemic verbal fluency; SVF: semantic verbal fluency; p values from Kruskal-Wallis H test

[a] Differences are significant after Bonferroni's post hoc test

[b] this value is constant.

had linked the bulbar onset with greater cognitive deficits [47, 48], the present study suggests that there are not differences regarding the verbal fluency compounds. However, a negative relationship was found between the motor alteration and the quantity and quality of the responses produced. Given that verbal fluency has been described as the most sensitive test to detect frontal alterations of ALS and FTD [49, 50], studying the underlying aspects of production was of interest. In line with previous studies [51–53], the dysexecutive component of the ALS-FTD complex was also reflected in the switches and clusters [25, 26].

A novelty was that this study used 2 minutes instead of the usual 1 minute in the study of verbal fluency. Because 'automatic' production occurs during the first 30 seconds of the task, the extension of time allows the voluntary activation search strategies to be studied [54]. This methodology allowed an overview analysis of the pathological motor component of the disease. Although there are ways to accommodate for the range of motor impairment, such as that proposed by Abrahams [55], some patients cannot write or say words quickly [56], and deficits can be exacerbated. The 2-minute task has also made it possible to notice some effects in the production pattern. For example, the PVF score was positively related to the number of perseverances. However, this pattern was reversed in the healthy population. It seems that those people who have a greater ability to generate words, based perhaps on a significantly higher number of years of education, were not exempt from executive errors that caused perseverance to occur. In contrast, the scarcity of words in some people may have not allowed this phenomenon to unfold and, it may have been more difficult to record for this reason. In this line, patients with more years of education produced significantly fewer intrusions. It has previously been pointed out that cognitive reserve plays a role as a protective factor against dementia in ALS [57, 58]. Continuing with the clinical factors, it has been observed that a significant relationship exists between motor alteration and verbal fluency: patients with more severe alterations produced significantly fewer words, particularly in the semantic task. According to Abrahams et al. [59], the participation of temporal regions can cause deficits in semantic processing, and also result in in confrontational naming deterioration. Therefore, it is not surprising that the ALS group in this study exhibited significant impairment in confrontational naming in.

It has been previously pointed out that altered processes of working memory, attention and inhibition could be responsible for the difficulty in satisfactorily performing the verbal fluency task [60, 61]. As these are alterations have been described previously in ALS, this finding does not come as a surprise. Some studies have suggested that the cognitive processes responsible for perseverations in SVF are based on the right hemisphere [62]. The atrophy of this hemisphere has been identified as a major biomarker in cognitive impairment in ALS [63–65].

Surprisingly, no statistically significant difference was observed between the levels of behavioural alteration observed in ALS. These seem to be relatively independence of the behavioural and cognitive measures in the sample evaluated; only people with severe behavioural alterations showed a significantly lower number of switches, and a smaller number of clusters in SVF. These findings are similar to those reported by Schmolck, Schulz and York [27]. It is believed that phonemic fluency invokes prefrontal and frontal functions due to the strategic processes required to search for the word [66, 67], whereas semantic fluency is usually located in the left anterior temporal lobe, where representations are classified by meaning [68]. This anatomical difference is also expressed in the results obtained here, where the score in SVF and clusters decrease in those patients with ALS that has been classified as having a severe alteration.

This study highlights the value of intrusions in the dysexecutive syndrome of ALS. As was seen previously, the patients without alterations in behaviour did not differ from the healthy population in the number of words produced, although they seemed to have a great tendency

to generate intrusions. Intrusions only occurred during the PVF task, but not during the SVF task. In our opinion, there is a greater dependence on the executive domain that triggered these errors. Another explanation could be the greater number of trials during the phonological task (three) in contrast to the semantics task (only one). It has been noted that memory is frequently altered in patients with ALS [69–72], and this alteration may cause a failure in the suppression of incorrect responses. Therefore, the very noticeable presence of intrusions during the evaluation of verbal fluency could be a sign to distinguish patients with ALS with severe behavioural impairment. Notably, the discriminant function analysis indicated that it is the variable that best differentiated people from ALS from healthy controls. Nevertheless, the great number of predictors involved in the analysis could affect negatively to the statistical power and forces this result to be considered with caution.

Some previous studies have pointed to the lack of association between the severity of dementia and the number of perseverations [73]. Taking into account the role of inhibition responses in perseverations [74, 75], a greater number of perseverations were expected. Some authors have suggested that inhibitory processes are necessary in verbal fluency tasks to suppress previously generated responses [60]. In this sense, the intuitive rule of not repeating a word can be understandable, in contrast to the explicit command not to use proper nouns. Therefore, this rule demands a greater load on cognitive resources than monitoring previous words. The inhibition response can also be important in the number of total words generated, as it may prevent strong responses and consider those words that are less likely to be produced [75]. Because of this, it is possible that the fewer words generated by the control group may be the consequence of a weaker executive functioning that produces repetitions. In this sense, although repetitive and persistent behaviour is a usual characteristic of people with FTD [76], it seems that this repetitive characteristic is independent of repetition (understood as an alteration dependent on executive processes), which could be more linked to processes involved in monitoring the task, and not so much to a disinhibition problem.

## Limitations

There are several limitations in this study that can be addressed in future research. Firstly, the evaluation of 2 minutes per letter may require considerable effort in ALS patients, especially those with significant bulbar or respiratory impairment. Secondly, this study did not include any complementary measure for testing executive functions and did not apply complementary measure for testing cognitive status in ALS as ECAS [77] nor other instrument which can be useful for distinguish patients cognitively impaired as MoCA or FAB [78]. Additionally, this study did not possess additional neuroimaging data that can support the findings. Thirdly, there were significant differences between the ALS and control groups in the Boston Naming Test result, so there may be specific language impairment in ALS patients. Future research should address these limitations, and also review those records in order to retrospectively study the importance of intrusions in the cognitive deterioration of patients with ALS.

## Supporting information

**S1 Raw data.**
(SAV)

## Author Contributions

**Conceptualization:** Manuel Perez, Oscar Martínez, Luis Varona.

**Data curation:** Maitane García.

**Investigation:** Manuel Perez.

**Methodology:** Mohammad Al-Rashaida, Paula Luna.

**Project administration:** Imanol Amayra, Esther Lazaro.

**Resources:** Manuel Perez, Oscar Martínez, Luis Varona.

**Supervision:** Patricia Caballero, Sarah Berrocoso, Juan Francisco López-Paz, Alicia Aurora Rodríguez.

**Writing – original draft:** Manuel Perez.

**Writing – review & editing:** Mohammad Al-Rashaida.

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
