## [Decision Letter · Decision Letter 0]

2 Mar 2020

PONE-D-19-20020

Intrusion errors during verbal fluency task in amyotrophic lateral sclerosis

PLOS ONE

Dear Dr. Manuel Perez,

Thank you for submitting your manuscript to PLOS ONE. After careful consideration, we feel that it has merit but does not fully meet PLOS ONE’s publication criteria as it currently stands. Therefore, we invite you to submit a revised version of the manuscript that addresses the points raised during the review process.

We would appreciate receiving your revised manuscript by Apr 10 2020 11:59PM. To enhance the reproducibility of your results, we recommend that if applicable you deposit your laboratory protocols in protocols.io, where a protocol can be assigned its own identifier (DOI) such that it can be cited independently in the future. For instructions see: http://journals.plos.org/plosone/s/submission-guidelines#loc-laboratory-protocols

We look forward to receiving your revised manuscript.

Kind regards,

Weidong Le

Academic Editor

PLOS ONE

Journal Requirements:

2. Please describe in your methods section how capacity to consent was determined for the participants in this study. For instance, please discuss how assessing participant's capacity to consent was discussed with the IRB, or whether caregivers were consulted to determine whether participants were able to consent.

a)    Please provide an amended Funding Statement that declares *all* the funding or sources of support received during this specific study (whether external or internal to your organization) as detailed online in our guide for authors at http://journals.plos.org/plosone/s/submit-now.  

b)    Please state what role the funders took in the study.  If any authors received a salary from any of your funders, please state which authors and which funder. If the funders had no role, please state: "The funders had no role in study design, data collection and analysis, decision to publish, or preparation of the manuscript."

Additional Editor Comments (if provided):

un-submitted comments from another reviewer (minor revision)

discussion, paragraph4: last sentence, "where the score in SVF and clusters decrease...moderate/severe alteration". It seem to be different with the results in Table 3.

discussion, paragraph5: "As was previously, the patients without alterations in behaviour did not differ from the healthy population in the number of ords produced...", but the results didn't shown this content.

discussion, paragraph5:"we suggest that patients with moderate and severe alterations did not differ in the number of intrusion because...". From Table3 result, the severe group didn't produce enough word(PVF score -1.15), but moderate group, PVF score was similar with none or mild group, so how to explain the moderate group did not differ in the number of intrusion?

result, last paragraph: "ELA"-"ALS"?

discussion, paragraph1:"ALS-FDT"-"ALS-FTD"

Reviewers' comments:

Reviewer's Responses to Questions

**Comments to the Author**

1. Is the manuscript technically sound, and do the data support the conclusions?

Reviewer #1: Yes

2. Has the statistical analysis been performed appropriately and rigorously? 

Reviewer #1: Yes

3. Have the authors made all data underlying the findings in their manuscript fully available?

Reviewer #1: Yes

4. Is the manuscript presented in an intelligible fashion and written in standard English?

Reviewer #1: Yes

5. Review Comments to the Author

Reviewer #1: Cognitive decline and deterioration of executive functions are common in amyotrophic lateral sclerosis (ALS). This study was conducted to evaluate the qualitative components of verbal fluency in patients with ALS. They found that ALS group produced a remarkably higher number of intrusions but not perseverations (independently from motor state or behavioral symptoms). The study is well done and addresses an interesting topic. I only have some minor comments and suggestions:

Introduction:

clearly written.

Methods:

How were controls recruited?

Exclusion/Inclusion: How were "critical levels of anxiety and/or depression at the time of the Evaluation" operationalized/assessed (questionnaire, cutoff ets)?

Results:

It seems that you had 12 predictors in the stepwise model. Given that you only had 2x42 persons these are too many independent variables and the statistical test is probably underpowered.

What is the "ELA"group. This abbreviation was not introduced before.

Table 1. How was disease progression calculated?

Table 2. Using Mann Whitney implies that data are not normally distributed. Thus, you might consider to report median + IQR instead of mean + SD.

Tables: please also Report the p-values for ns.

For Group comparisons please also Report the 95%CI

Did you observe any relationship between bulbar involvement (and consecutive higher disease progression and more frontal involvement) and dysexecutive components?

Discussion:

A major limitation is the missing assessment of general cognitive function (eg ECAS, MoCa). Please comment on this.

6. PLOS authors have the option to publish the peer review history of their article (what does this mean?). If published, this will include your full peer review and any attached files.

Reviewer #1: No

---

## [Author Response · Author response to Decision Letter 0]

9 Apr 2020

Reviewer#1

discussion, paragraph4: last sentence, "where the score in SVF and clusters decrease...moderate/severe alteration". It seem to be different with the results in Table 3.

Indeed, there was a mistake in the original manuscript and the text of paragraph 4 has been amended. 

discussion, paragraph5: "As was previously, the patients without alterations in behaviour did not differ from the healthy population in the number of ords produced...", but the results didn't shown this content.

The analysis was performed but not included in the results section. Now it has been included and indeed, the Mann-Whitney U analysis reveals that there are not significant differences between those groups. 

discussion, paragraph5:"we suggest that patients with moderate and severe alterations did not differ in the number of intrusion because...". From Table3 result, the severe group didn't produce enough word(PVF score -1.15), but moderate group, PVF score was similar with none or mild group, so how to explain the moderate group did not differ in the number of intrusion?

After discussing this particular we have decided to delete it due to lack of theoretical fundamentals of our explanations. 

result, last paragraph: "ELA"-"ALS"?

From native «Esclerosis Lateral Amiotrófica». The terms was replaced by ALS in the manuscript.

discussion, paragraph1:"ALS-FDT"-"ALS-FTD"

The terms was replaced by ALS-FTD in the manuscript.

Reviewer#2

How were controls recruited?

Control group participants were recruited through public advisements in hospitals. This has been included in the original manuscript. 

Exclusion/Inclusion: How were "critical levels of anxiety and/or depression at the time of the Evaluation" operationalized/assessed (questionnaire, cutoff ets)?

Critical levels of anxiety and/or depression were based on the HADS scores and its cutoff score stablished by the test ( ≥ 8 for the identification of suspicious cases and ≥ 11 for safe cases on both subscales of anxiety and depression). This particular has been included in the corrected manuscript. 

Results:

It seems that you had 12 predictors in the stepwise model. Given that you only had 2x42 persons these are too many independent variables and the statistical test is probably underpowered.

We strongly agree with this point. Therefore, despite the clinical importance that it may have, this fact has been included as another limitation of the study.

What is the "ELA"group. This abbreviation was not introduced before.

From native «Esclerosis Lateral Amiotrófica». The terms was replaced by ALS in the manuscript.

Table 1. How was disease progression calculated?

The rate of disease progression was proposed by Kimura et al. (2006) and is calculated as follows: (48 – ALSFRS-R at time of diagnosis) / duration from onset to diagnosis (months) 

This formula has been included in the original manuscript and the reference in the list. 

Table 2. Using Mann Whitney implies that data are not normally distributed. Thus, you might consider to report median + IQR instead of mean + SD.

We fully agree with that statement and after the corrections, Tables are expressed in median and IQR.

Tables: please also Report the p-values for ns.

Non-significant P values are reflected in the new tables.

For Group comparisons please also Report the 95%CI

The 95%CI data is reflected in the new tables.

Did you observe any relationship between bulbar involvement (and consecutive higher disease progression and more frontal involvement) and dysexecutive components?

We had considered this fact from the beginning and after carrying out the analyzes regarding the type of disease onset (ie. bulbar vs. spinal) we have not found significant differences. However, due to its clinical importance question, we have included this in the Results and Discussion sections. Different studies have been cited and referenced in the manuscript.

Discussion:

A major limitation is the missing assessment of general cognitive function (eg ECAS, MoCa). Please comment on this.

We fully agree with the mistake of not using screening tests. In the corrected manuscript, their applications and virtues have been discussed in the Limitations section. Different studies have been cited and referenced in the manuscript.

---

## [Decision Letter · Decision Letter 1]

16 Apr 2020

PONE-D-19-20020R1

Intrusion errors during verbal fluency task in amyotrophic lateral sclerosis

PLOS ONE

Dear Dr. Manuel Perez,

Thank you for submitting your manuscript to PLOS ONE. After careful consideration, we feel that it has merit but does not fully meet PLOS ONE’s publication criteria as it currently stands. Therefore, we invite you to submit a revised version of the manuscript that addresses the points raised during the review process.

We would appreciate receiving your revised manuscript by May 31 2020 11:59PM. To enhance the reproducibility of your results, we recommend that if applicable you deposit your laboratory protocols in protocols.io, where a protocol can be assigned its own identifier (DOI) such that it can be cited independently in the future. For instructions see: http://journals.plos.org/plosone/s/submission-guidelines#loc-laboratory-protocols

We look forward to receiving your revised manuscript.

Kind regards,

Weidong Le

Academic Editor

PLOS ONE

Reviewers' comments:

Reviewer's Responses to Questions

**Comments to the Author**

1. If the authors have adequately addressed your comments raised in a previous round of review and you feel that this manuscript is now acceptable for publication, you may indicate that here to bypass the “Comments to the Author” section, enter your conflict of interest statement in the “Confidential to Editor” section, and submit your "Accept" recommendation.

Reviewer #1: (No Response)

2. Is the manuscript technically sound, and do the data support the conclusions?

Reviewer #1: Yes

3. Has the statistical analysis been performed appropriately and rigorously? 

Reviewer #1: I Don't Know

4. Have the authors made all data underlying the findings in their manuscript fully available?

Reviewer #1: Yes

5. Is the manuscript presented in an intelligible fashion and written in standard English?

Reviewer #1: Yes

6. Review Comments to the Author

Reviewer #1: All my concerns were addressed.

Three minor aspects remain:

- It is more common to report ´median (IQR)´instead of ´IQR (median)´

- Now the authors state that their statistic is probable underpowered. I do not know if this is the right way. I am not a statistician, but I would prefer to use appropriate statistics for the sample size. I recommend to involve (statistical) editor or statistician to solve this issue.

- I recommend to translate the raw data into English.

7. PLOS authors have the option to publish the peer review history of their article (what does this mean?). If published, this will include your full peer review and any attached files.

Reviewer #1: No

---

## [Author Response · Author response to Decision Letter 1]

4 May 2020

Dear Editors and Reviewers:

Thank you for your letter and for the reviewers’ comments concerning our manuscript. Those comments are very helpful for revising and improving our paper. We have studied comments carefully and have made correction which we hope meet with approval.

---

## [Editor Report · Decision Letter 2]

5 May 2020

Intrusion errors during verbal fluency task in amyotrophic lateral sclerosis

PONE-D-19-20020R2

Dear Dr. Manuel Perez,

We are pleased to inform you that your manuscript has been judged scientifically suitable for publication and will be formally accepted for publication once it complies with all outstanding technical requirements.

With kind regards,

Weidong Le

Academic Editor

PLOS ONE
---

## [Editor Report · Acceptance letter]

20 May 2020

PONE-D-19-20020R2 

Intrusion errors during verbal fluency task in amyotrophic lateral sclerosis 

Dear Dr. Perez:

I am pleased to inform you that your manuscript has been deemed suitable for publication in PLOS ONE. Congratulations! Your manuscript is now with our production department. 

With kind regards,

on behalf of

Dr. Weidong Le 

Academic Editor

PLOS ONE